# Estimation of Water Stress in Potato Plants Using Hyperspectral Imagery and Machine Learning Algorithms

**Julio Martin Duarte-Carvajalino** [1,*], **Elías Alexander Silva-Arero** [1], **Gerardo Antonio Góez-Vinasco** [1], **Laura Marcela Torres-Delgado** [2], **Oscar Dubán Ocampo-Paez** [2] and **Angela María Castaño-Marín** [1]

1. Corporación Colombiana de Investigación Agropecuaria—Agrosavia,
   Centro de Investigación Tibaitatá—Km 14 Vía Mosquera, Bogotá D.C. 250047, Cundinamarca, Colombia;
   esilva@agrosavia.co (E.A.S.-A.); ggoez@agrosavia.co (G.A.G.-V.); amcastano@agrosavia.co (A.M.C.-M.)
2. Universidad Nacional de Colombia, Carrera 45 N° 26-85–Edificio Uriel Gutiérrez,
   Bogotá D.C. 111321, Cundinamarca, Colombia; lamtorresde@unal.edu.co (L.M.T.-D.);
   odocampop@unal.edu.co (O.D.O.-P.)
* Correspondence: jmduarte@agrosavia.co; Tel.: +57-3132495914

**Abstract:** This work presents quantitative detection of water stress and estimation of the water stress level: none, light, moderate, and severe on potato crops. We use hyperspectral imagery and state of the art machine learning algorithms: random decision forest, multilayer perceptron, convolutional neural networks, support vector machines, extreme gradient boost, and AdaBoost. The detection and estimation of water stress in potato crops is carried out on two different phenological stages of the plants: tubers differentiation and maximum tuberization. The machine learning algorithms are trained with a small subset of each hyperspectral image corresponding to the plant canopy. The results are improved using majority voting to classify all the canopy pixels in the hyperspectral images. The results indicate that both detection of water stress and estimation of the level of water stress can be obtained with good accuracy, improved further by majority voting. The importance of each band of the hyperspectral images in the classification of the images is assessed by random forest and extreme gradient boost, which are the machine learning algorithms that perform best overall on both phenological stages and detection and estimation of water stress in potato crops.

**Keywords:** water stress; potato; hyperspectral image; machine learning; band importance

## 1. Introduction

Potato (*Solanum tuberosum* L.) is the third most important food crop in the world [1]. The potato provides an economic and rich source of carbohydrates and it is included in the diet of both developed and undeveloped countries. Water deficit is the most important abiotic stress affecting the development, productivity, and quality of potato cultivars [2]. Hence, it is important to detect, as early as possible, signs of water stress in potato plants avoiding production and quality losses. Due to climate change, crops worldwide are suffering from unexpected and longer severe weather changes such as droughts, which are becoming increasingly more intense [3]. Specifically in Colombia, a good portion of areas suitable for potato production are vulnerable to increased aridity, soil erosion, desertification, and variations in the hydrological system as a consequence of climate change [4]. Therefore, there is a need to map water stress in potato crops using non-destructive technologies such as remote sensing.

Recently, a spectroradiometer (350–2500 nm) was used to explore the effect of water stress on the spectral reflectance of bermudagrass and five vegetation indexes were studied [5]. In the case of potato crops, 12 vegetation indexes including four Normalized Water Indexes (NWIs), have been studied to detect water stress in potato leaves under different watering conditions using also a spectroradiometer (350–2500 nm) [4]. The results indicate clear differences in the spectrum of water-stressed leaves in the 700–1300 nm range [4].

Remote sensing technologies using unmanned aerial vehicles (UAVs) acquiring visible and thermal images were used to map water stress in barley crops [6]. The detection of water stress in plants using aerial imagery has focused on thermal imagery to estimate plant temperature relative to the air temperature computing NWIs. Since stomata close under water stress, the temperature of the leaves relative to the air increases [6–8]. More recently, remote sensing imaging technologies using visible, near-infrared (NIR), short wave infrared (SWIR), and thermography have been proposed to detect water stress in potato crops [9]. Rather than using broadband multispectral images, hyperspectral imagery and machine learning algorithms have been proposed to determine the quality of food products [10]. Hyperspectral imagery (400–1000 nm) has also been proposed to detect water stress in potato crops using spectral indexes [11]. Hyperspectral imagery (400–2500 nm) was used in combination with partial least squares–discriminant analysis (PLS-DA) and partial least squares–support vector machine (PLS-SVM) classification to detect abiotic and biotic drought stress in tomato canopies [12]. Hyperspectral imagery (450–1000 nm) in combination with machine learning algorithms (random forest and extreme gradient boost) has been also used to detect water stress in vine canopies [13]. Another possibility for detecting water stress in plants is to use radar remote sensing technologies [14,15] with the advantage of penetrating the clouds, a limitation of visible and thermal imagery. Finally, ultrasound wave spectroscopy has also been used to estimate the water content of plant leaves using convolutional neural networks and random forest algorithms [16].

As previously indicated, work on detecting water stress in potato cultivars has been based on vegetation indexes (*NDVI*, the Simple Ratio, the Photochemical Reflectance Index, the pigment-specific simple ratio of Chlorophyll-a, the reflectance water index, the Normalized Water Indexes and the dry Zea N index). Here we use a hyperspectral camera (400–1000 nm) and several well-known machine learning algorithms to detect water stress in potato hyperspectral images and to estimate the degree of water stress: none, light, moderate and severe, using all images bands. The use of machine learning algorithms allows us to determine which regions in the spectral signature of the leaves are more influential to better estimate water stress from remote sensing using images in the visible (400–700 nm) and near-infrared (*NIR*) (700–1000 nm) bands.

## 2. Material and Methods

### 2.1. Plant Material and Experimental Design

The experiment was developed in greenhouse number 17 of AGROSAVIA (Corporación Colombiana de Investigación Agropecuaria), Tibaitatá research center, Colombia (4°41′25.7064″ N, 74°12′08.23″ W) at 2543 m above the sea level. Certified seeds of *Solanum tuberosum* L., variety Diacol Capiro were planted in the greenhouse. The experiment consisted of a randomized complete blocks design in a factorial 2 × 4 arrangement. The first factor considered was the level of plant development (phenological stage), this was fixed according to [17]: tubers differentiation (TD) and maximum tuberization (MT) (Appendix A). The second factor was the level of water stress severity, determined by the hydric potential of the leaves, measured using a Scholander pressure chamber in Mega Pascals (Mpa). Control plants have a hydric potential in the 0–−0.49 Mpa range, light (L) water stress has a hydric potential in the −0.5–−0.59 Mpa range, moderate (M) water stress has a hydric potential in the −0.6–−0.89 Mpa, and severe (S) water stress has a hydric potential equal to or lower than −0.9 Mpa. These hydric potential ranges were selected based on [18,19], and previous research experience of AGROSAVIA in greenhouses containing potato crops.

Potato plants were sown in a greenhouse in a loamy soil that was kept at field capacity (soil water potential did not decline below −0.033 MPa) by drip irrigation from sowing until the 9th and 13th week after sowing, when each stage of development was reached (TD and TF, respectively). At that time, the water supply was suspended, and the water potential in the leaf was measured daily until reaching each level of stress (L, M, S). Control plants had a water supply throughout the experiment.

### 2.2. Hyperspectral Imagery

The hyperspectral images were acquired using a 710-VP Surface Optics Corporation camera with 520 × 696 pixels and 128 spectral bands in the 400–1000 nm range, using the Environment for Visualizing Images (ENVI) format. The images were taken at 3 m above the plant's canopy level and the camera looking downwards. The image acquisition campaigns were done at around the same hour of the day. Figure 1 shows a false-color image of the canopy of a plant loaded and visualized with MultiSpec [20]. As can be seen from this image a Spectralon reflectance white panel is also used on each image to convert the hyperspectral intensity images to reflectance. It is easy to segment the white Spectralon panel from the hyperspectral image by computing the average of the red, green, blue, and *NIR* bands and dividing that image by the maximum intensity. Figure 2 shows this normalized average, where the Spectralon reflectance panel can be segmented from the image using a threshold above 0.5.

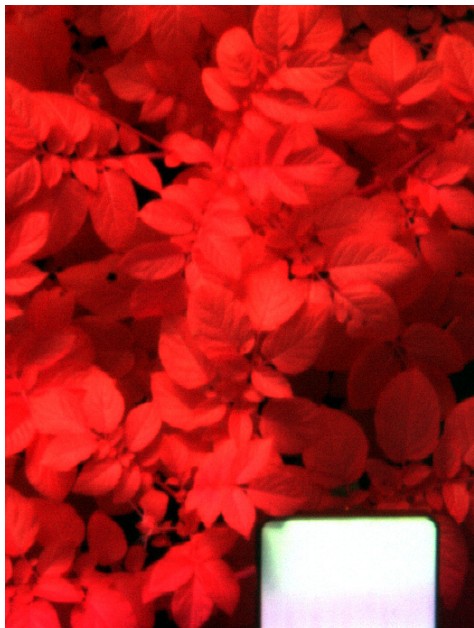

**Figure 1.** Hyperspectral image taken at 3 m above the canopy. False-color image showing red as band 90, green as band 60, and blue as band 40.

The reflectance of each hyperspectral image can be computed using:

$$\rho(x, y, \lambda) = \frac{I(x, y, \lambda)\rho_S(\lambda)}{Is(\lambda)} \tag{1}$$

where $\rho(x, y, \lambda)$ is the reflectance image at pixel coordinates $x, y$ and waveband $\lambda$, $I(x, y, \lambda)$ is the raw intensity image at pixel coordinates $x, y$, and waveband $\lambda$, $\rho_S(\lambda)$ the known reflectance of the Spectralon panel at $\lambda$ wavelength (0.99 at visible and *NIR* ranges) and $Is(\lambda)$ the mean intensity of the Spectralon panel at waveband $\lambda$. Once the hyperspectral images are converted to reflectance, it is necessary to segment the canopy from its background. The Normalized Difference Vegetation Index (*NDVI*) has widely been used to detect vegetation canopy [21]:

$$NDVI = \frac{\rho_{NIR} - \rho_{red}}{\rho_{NIR} + \rho_{red}} \tag{2}$$

where $\rho_{NIR}, \rho_{red}$ are the reflectances at the *NIR* and red wavelengths, respectively. However, the *NDVI* is affected by several factors including shadows [21] that could lead to 0/0

undefined values. To avoid this, we used the Soil-Adjusted Vegetation Index (*SAVI*) that overcomes the issues of the *NDVI* [21] and selected those values where *SAVI* > 0.3 (Figure 3):

$$SAVI = 1.5 \frac{\rho_{NIR} - \rho_{red}}{(0.5 + \rho_{NIR} + \rho_{red})} \tag{3}$$

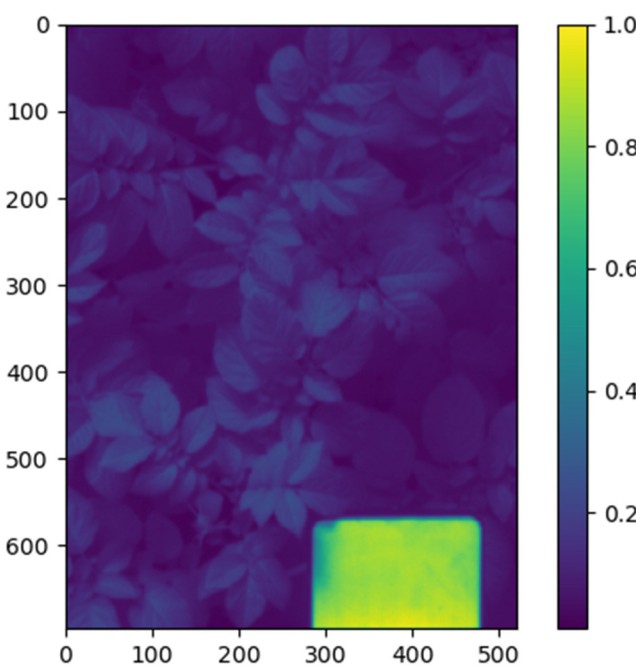

**Figure 2.** Normalized sum of red, green, blue, *NIR* bands.

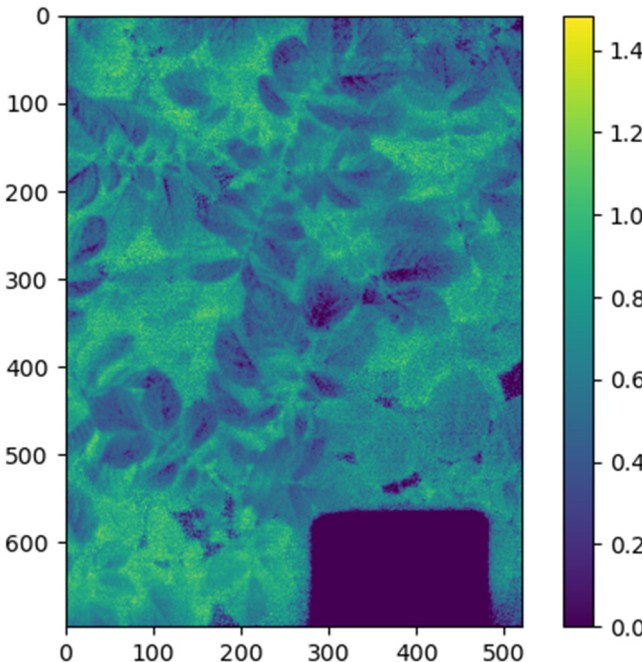

**Figure 3.** *SAVI* greater than 0.3 to detect leaves.

From the image campaign at the tubers differentiation phenological stage, 64 images were acquired to be used for the machine learning algorithms (stressed and control plants) with water stresses that range from 3 to 20 days. From the image campaign at the maximum

tuberization phenological stage, 52 images were acquired to be used for the machine learning algorithms (stressed and control plants) with water stresses that range from zero to nine days. The reading and preprocessing of the hyperspectral images were done using Python 3.8.5 that comes with Anaconda [22]. The Python spectral library [23] was used to read the hyperspectral images.

There are control plants that provide images for the control class and there are several images for each stress condition, taken at different days after the application of each stress level.

### 2.3. Machine Learning Algorithms

Two supervised classification tasks for the two phenological stages of the potato crops were carried out: detection of water stress i.e., the plant is water-stressed or not (two classes) and the estimation of the water level of stress i.e., the plant is not water-stressed, is lightly water-stressed, is moderately stressed or severely stressed (four classes). To perform these classification tasks six well-known machine learning algorithms were used:

- Random decision forest (RF) [24] using 100 trees, with a balanced class weight. RF are an ensemble of decision trees, the class predicted corresponds to the class most voted for the decision trees.
- Multi-layer perceptron (MLP) [25] with an input layer having equal nodes as the number of bands (128) and an output layer having equal nodes as the number of classes (2 or 4). Each layer is followed by a batch normalization layer [26], a dropout layer [27] with a probability of 0.2, a rectified linear activation function (RELU, a function that will output the same input if it is positive, zero otherwise) [28] on the input layer, and a Softmax activation function [28] on the output layer for the case of four classes or a Sigmoid activation function [28] for the case of two classes classification (see Figure 5). An MLP neural network consists of layers of nodes: an input layer, hidden layers and an output layer. Except for the input nodes, each node is a neuron that uses a nonlinear activation function. Each node on a layer connects with each node of the following layer by a weight function. The neural network learns the weights from the training data.
- Convolutional neural networks (CNN) [29] with two convolutional layers using a kernel size of 3 and 20 filters each one. The two convolutional layers are followed by a batch normalization layer, a dropout (0.2) layer, and a RELU layer. After the two convolutional layers, a flatten layer follows to flatten out the last convolutional layer into MLP nodes. After the flatten layer, an input MLP layer of size equal to the half of nodes of the flatten layer follows, then a middle MLP layer with half the nodes of the previous layer and an output layer with equal nodes as the number of classes. Each MLP layer is followed by a batch normalization layer, a dropout (0.2) layer, and a RELU layer for the case of the first MLP layer and the second MLP layer. The last MLP layer is followed by a Softmax activation function in the case of four classes or a Sigmoid activation function, in the case of two classes (Figure 4). Convolutional neural networks are a kind deep learning neural network specialized on images, with convolutional layers applying different kinds of filters on patches of the images and then on previous convolutional layers, to capture variabilities at higher scales.
- Support vector machine (SVM) [30] using linear SVM with default parameters. SVM maps training examples to points in space so as to maximize the width of the gap between the classes.
- Extreme gradient boost (XGBoost) [31] using tree classifiers (gbtree) as weak learners and 100 estimators. Gradient boosting produces an ensemble of weak predictions (usually trees) models and generalizes them by the optimization of a differentiable loss function. XGBoost in an implementation of gradient boosting that uses a more regularized model formalization to control overfitting.
- AdaBoost (AB) [32] with 100 estimators. An AdaBoost classifier works by fitting a classifier that first fits the dataset and then fits additional copies of the classifier, but

giving more weight to the incorrectly classified instances, so subsequent classifiers focus on harder cases.

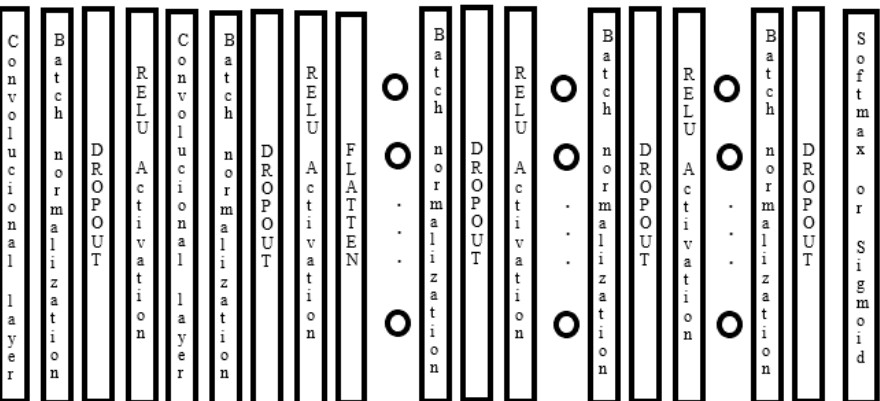

**Figure 4.** Convolutional neural network layout.

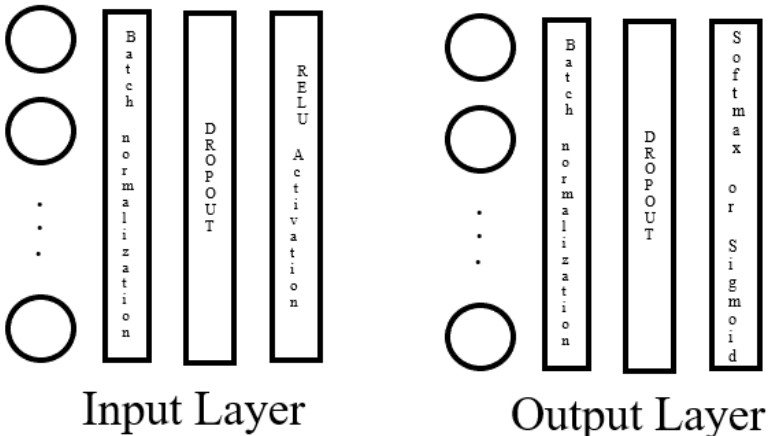

**Figure 5.** Multi-layer perceptron layout.

The RF, SVM, AB classifiers were implemented in Python 3.8 using the sklearn library. The MLP and CNN were implemented in Python 3.5 using the keras library with tensorflow under the hood in the High Performance Computing servers of Agrosavia, given the memory required by CNN. The XGBoost classifier was implemented using xgboost python library in Python 3.8.

Given the size of the images ($520 \times 696 \times 128$) and equipment memory constraints and processing times, only 10000 pixels were selected at random from the canopy (identified using $SAVI > 0.3$) on each image to train the classifiers forming a training dataset. In the case of CNN, a window of size $5 \times 5 \times 128$ was selected centered on each one of the 10,000 pixels selected at random in the canopy to form the CNN dataset. To evaluate the classifiers, five-fold cross-validation was employed to measure the probability of classfication overfitting, due to the tendency of classifiers to overfit the training dataset. Here, 80% of the dataset is used for training and 20% for testing the classifiers on each one of the five-fold cross-validation runs. In the case of MLP and CNN, 20% of the 80% available data for training is used for validation in such a way that the MLP or CNN models are saved only if the computed loss improves for the validation data, as an extra measure to avoid overfitting the dataset. Furthermore, the classifiers were trained with the full training dataset and then used to classify the whole canopy on each image (containing many more pixels unseen by the classifiers) using majority voting, i.e., selecting the class most pixels are classified with.

## 3. Results

Figure 6 shows the classification performance using two classes (water stress or control) for the phenological stage tubers differentiation using overall accuracy, sensitivity, and specificity (see confusion matrices in the Appendix B), where the standard deviation of the mean is indicated for accuracy, sensitivity, and specificity, as error bars. As can be seen from these results RF and XGBoost achieve the best classification performance, being XGBoost the best.

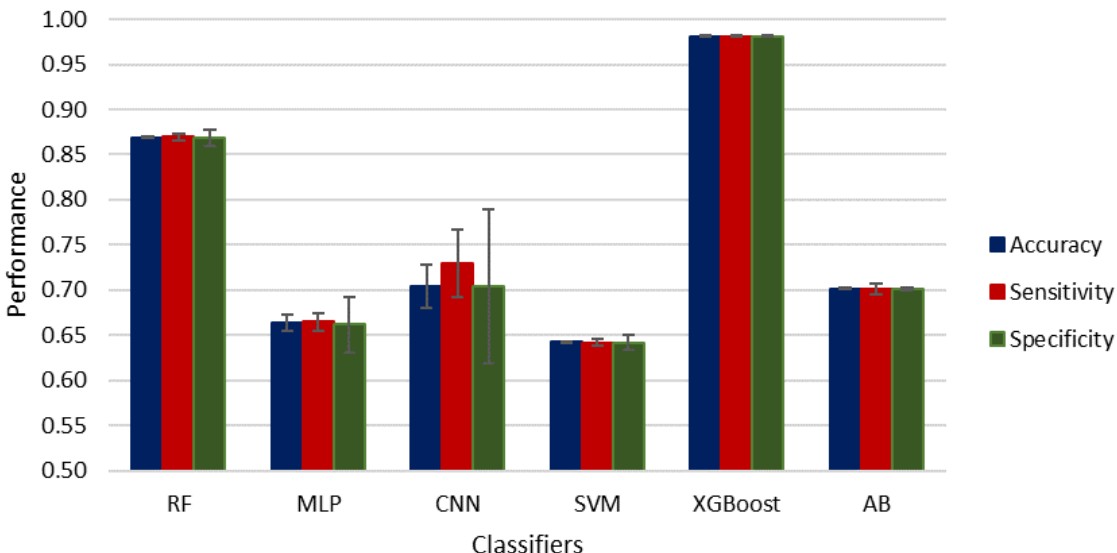

**Figure 6.** Classification performance, tubers differentiation phenological stage using two classes.

Table 1 compares the classification performance using the best three classifiers found: RF, XGBoost, and CNN alone and using Majority Voting (MV). This table shows that both RF and XGBoost correctly classify all the images using majority voting, followed by CNN.

**Table 1.** Comparison of classification performance of RF, XGBoost, and CNN alone and using MV for tubers differentiation phenological stage using two classes.

|  | RF | RF + MV | XGBoost | XGBoost + MV | CNN | CNN + MV |
|---|---|---|---|---|---|---|
| Accuracy | 0.8691875 | 1 | 0.98120469 | 1 | 0.69395156 | 0.875 |
| Sensitivity | 0.86965626 | 1 | 0.98114434 | 1 | 0.71711594 | 0.875855327 |
| Specificity | 0.86857087 | 1 | 0.98123905 | 1 | 0.69385401 | 0.875855327 |

Figure 7 shows the classification performance for the tubers differentiation phenological stage and four classes: control and three levels of water stress: light, moderate, and severe (see confusion matrices in the Appendix B), where the standard deviation of the mean is indicated for accuracy, sensitivity, and specificity, as error bars. In this case, XGBoost performs best, followed by RF and MLP. Table 2 compares the classification performance of the three best classifiers: RF, XGBoost, and CNN alone and using MV. In this case, XGBoost performs best, followed by RF and CNN.

**Table 2.** Comparison of classification performance of RF, MLP, and CNN alone and using MV for tubers differentiation phenological stage using two classes.

|  | RF | RF + MV | XGBoost | XGBoost + MV | CNN | CNN + MV |
|---|---|---|---|---|---|---|
| Accuracy | 0.811439063 | 0.90625 | 0.985457813 | 1 | 0.63530625 | 0.703125 |
| Sensitivity | 0.879199269 | 0.961538462 | 0.991209678 | 1 | 0.591587693 | 0.66889881 |
| Specificity | 0.707431992 | 0.829861111 | 0.978625726 | 1 | 0.495725174 | 0.517834596 |

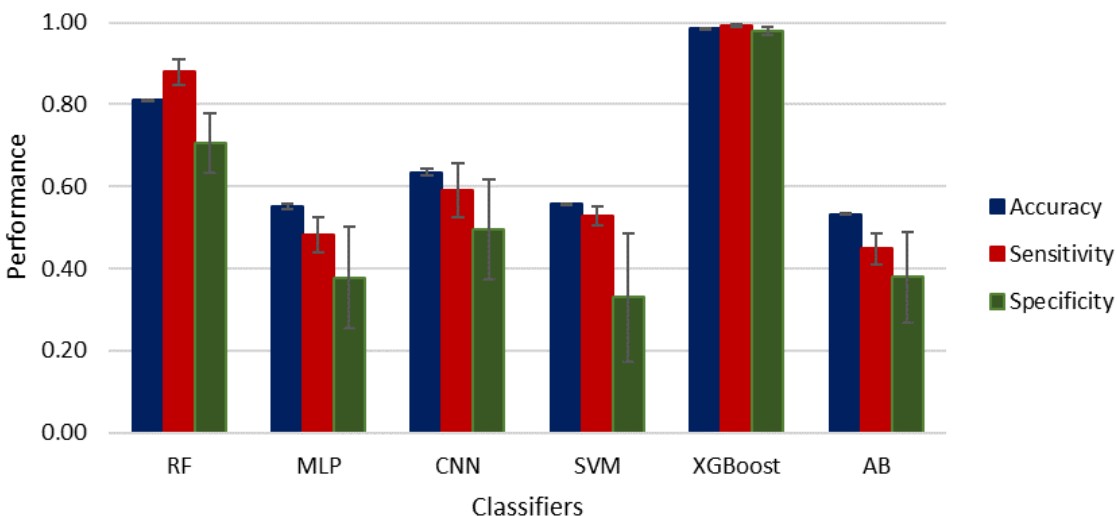

**Figure 7.** Classification performance, tubers differentiation phenological stage using four classes.

Figure 8 shows the classification performance at the maximum tuberization phenological stage using two classes: control and water stress (see confusion matrices in the Appendix B), where the standard deviation of the mean is indicated for accuracy, sensitivity, and specificity, as error bars. The best classifiers are XGBoost followed by RF and CNN. Table 3 compares the classification performance of RF, XGBoost, and CNN alone and using MV over all the images. This table shows RF and XGBoost both achieve perfect classification using MV of all the images taken at this phenological stage.

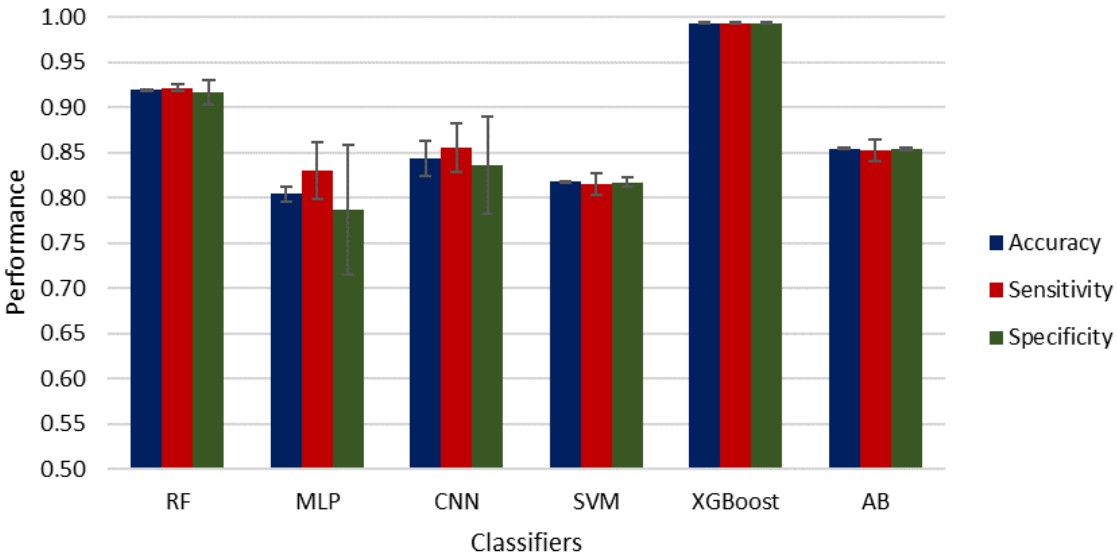

**Figure 8.** Classification performance, maximum tuberization phenological stage using two classes.

**Table 3.** Comparison of classification performance of RF, XGBoost, and CNN alone and using MV for tubers differentiation phenological stage using two classes.

|  | RF | RF + MV | XGboost | XGBoost + MV | CNN | CNN + MV |
|---|---|---|---|---|---|---|
| Accuracy | 0.92025577 | 1 | 0.99373077 | 1 | 0.84420769 | 0.980769231 |
| Sensitivity | 0.92156596 | 1 | 0.99353142 | 1 | 0.8555923 | 0.979166667 |
| Specificity | 0.91676559 | 1 | 0.99376627 | 1 | 0.83643118 | 0.982758621 |

Figure 9 shows the classification performance at the maximum rate of tubers phenological stage using four classes: control, light, moderate, and severe water stress (see confusion matrices in the Appendix B), where the standard deviation of the mean is indicated for accuracy, sensitivity, and specificity, as error bars. Here, XGBoost obtains the best performance, followed by RF and CNN. As in the case of the two classes, the classification accuracies are good and allow estimation of the water stress from the first day. Table 4 compares the classification performance of RF, XGBoost, and CNN alone and using MV, where it can be noticed that XGBoost in combination with MV achieves perfect classification, followed by RF and CNN.

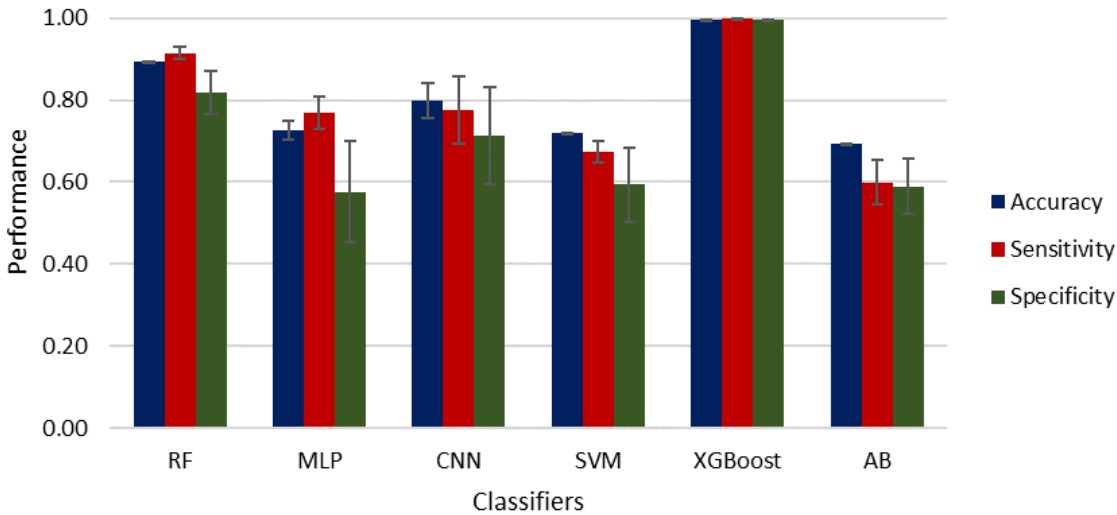

**Figure 9.** Classification performance, maximum tuberization phenological stage using four classes.

**Table 4.** Comparison of classification performance of RF, XGBoost, and CNN alone and using MV for the maximum tuberization phenological stage using four classes.

|  | RF | RF + MV | XGBoost | XGBoost + MV | CNN | CNN + MV |
|---|---|---|---|---|---|---|
| Accuracy | 0.894794231 | 0.980769231 | 0.997425 | 1 | 0.797767308 | 0.961538462 |
| Sensitivity | 0.914904761 | 0.991666667 | 0.997991496 | 1 | 0.775300134 | 0.964285714 |
| Specificity | 0.818412336 | 0.964285714 | 0.996019078 | 1 | 0.713138567 | 0.982758621 |

Figure 10 shows XGBoost classification results on some images of the tubers differentiation phenological stage using four classes. The color code here is green for no water stress, blue for light stress, yellow for moderate stress, and red for severe stress. Figure 10a shows the classification for a control plant (no water stress). Figure 10b shows a plant that suffered light stress. Figure 10c shows a plant that suffered moderate stress. Figure 10d shows a plant that suffered severe stress.

Figure 11 shows some XGBoost classification results for the maximum tuberization phenological stage using the same color code as in Figure 10.

Figure 12 shows the band importance for RF classification in the detection (two classes) and estimation (four classes) of water stress at the phenological stage of tubers differentiation. Figure 13 shows the same band importance for RF classification of two and four classes at the phenological stage of the maximum tuberization. As indicated in Figures 12 and 13 the most important bands for classification in RF are the violet, the red edge, and a few wavelengths in the NIR.

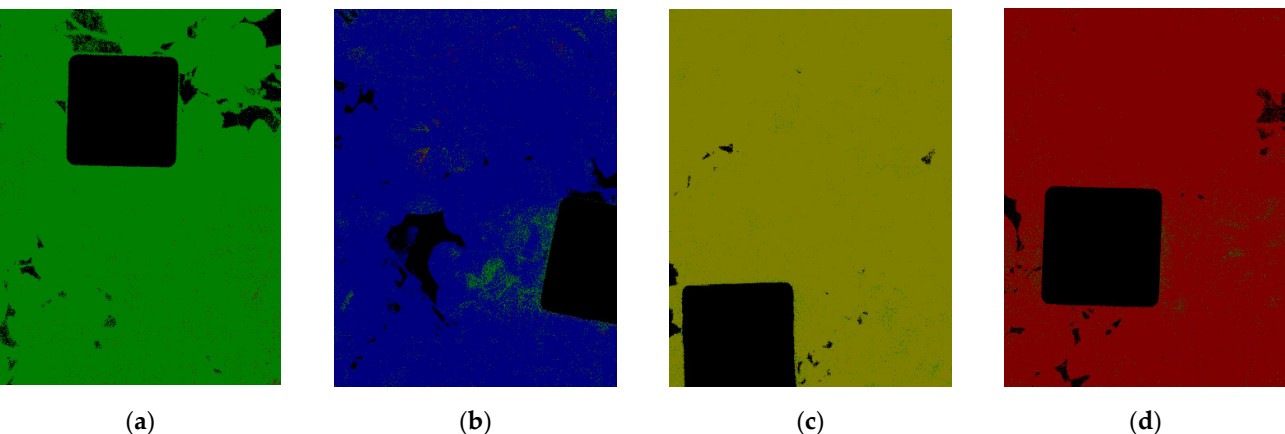

(**a**)        (**b**)        (**c**)        (**d**)

**Figure 10.** XGBoost classification of (**a**) image of a control plant, (**b**) image of a plant with light stress, (**c**) image of a plant with moderate stress, (**d**) image of a plant with severe stress.

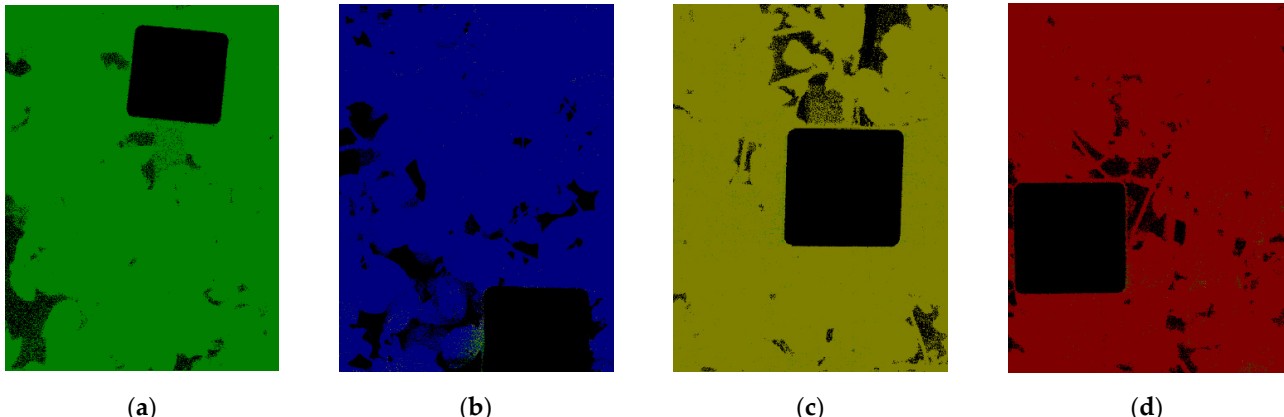

(**a**)        (**b**)        (**c**)        (**d**)

**Figure 11.** XGBoost classification of (**a**) control plant, (**b**) plant with light stress, (**c**) plant with moderate stress, (**d**) plant with severe stress.

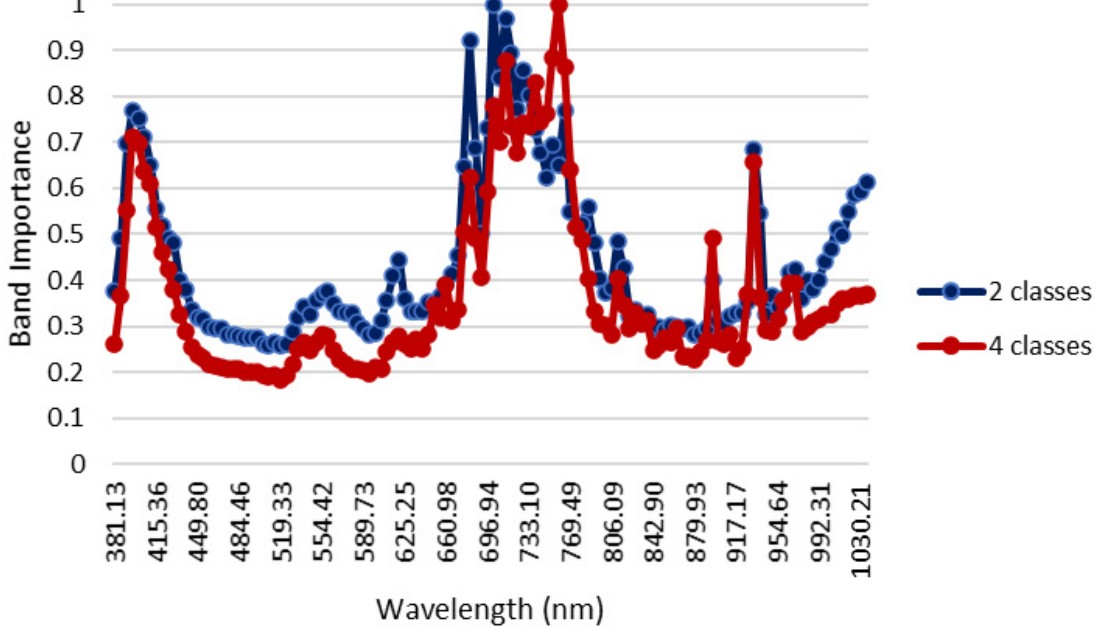

**Figure 12.** Band importance determined by RF on the tubers differentiation phenological stage.

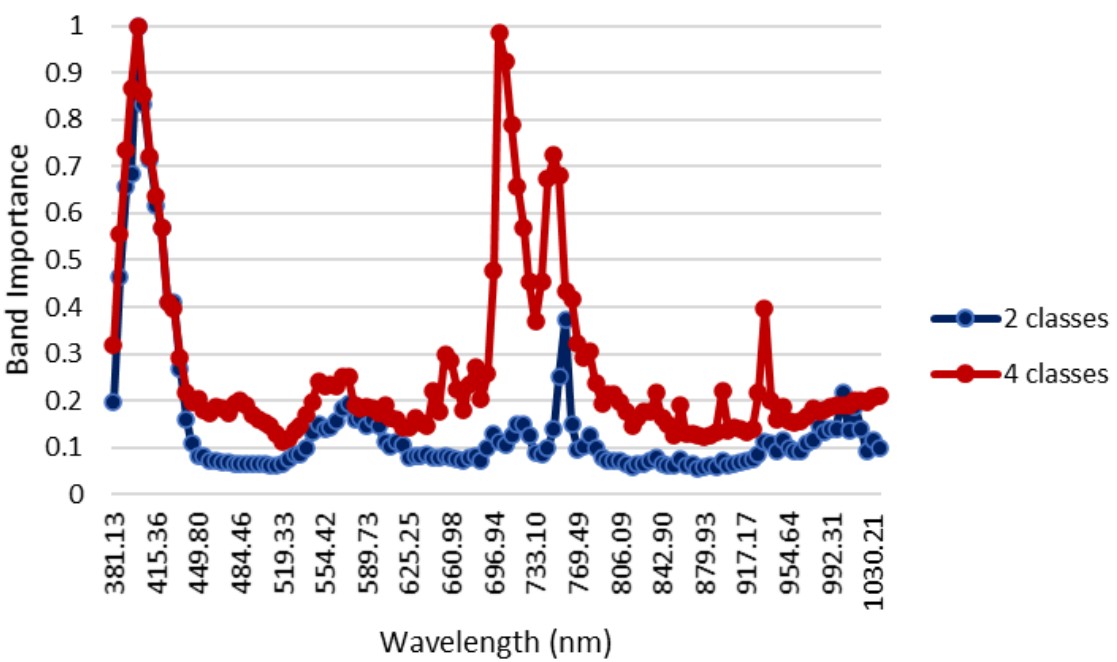

**Figure 13.** Band importance determined by RF on the maximum tuberization phenological stage.

Figure 14 shows the band importance for XGBoost classification in the detection (two classes) and estimation (four classes) of water stress at the phenological stage of tubers differentiation. Figure 15 shows the same band importance for XGBoost classification of two and four classes at the phenological stage of the maximum tuberization. From these figures, XGBoost considers important more bands than RF, i.e., it exploits better the spectral signature of the hyperspectral images. Band importance could help us identify which bands are better suited to detect water stress from multispectral imagery or to define water stress indices specially designed for potato crops.

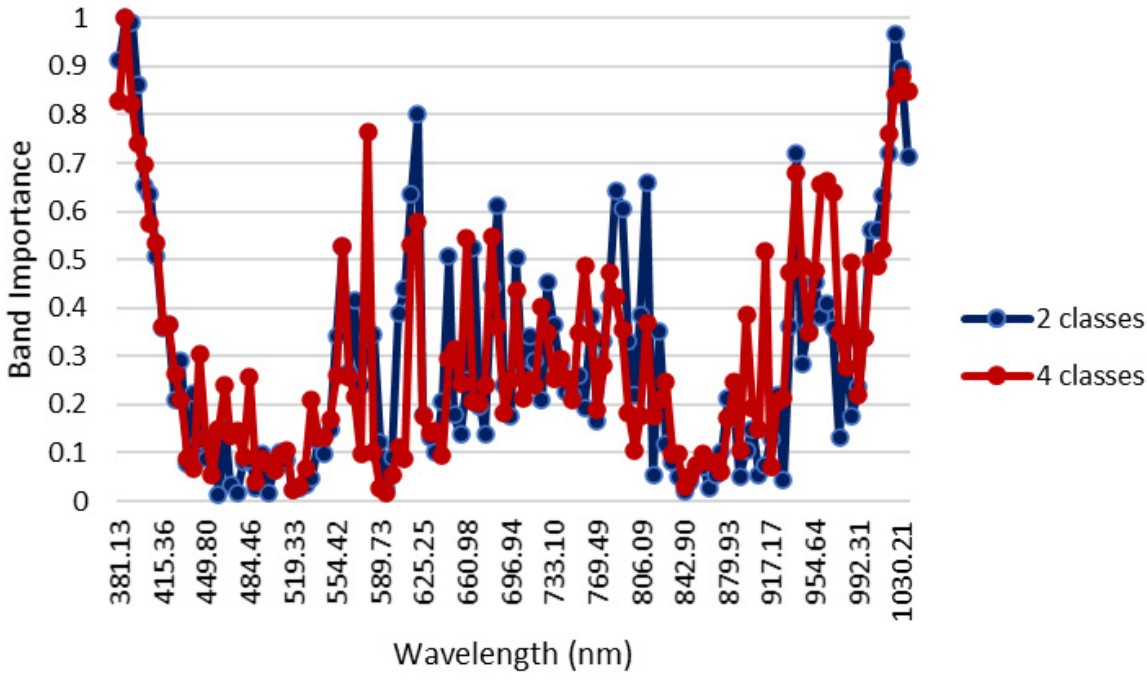

**Figure 14.** Band importance determined by XGBoost on the tubers differentiation phenological stage.

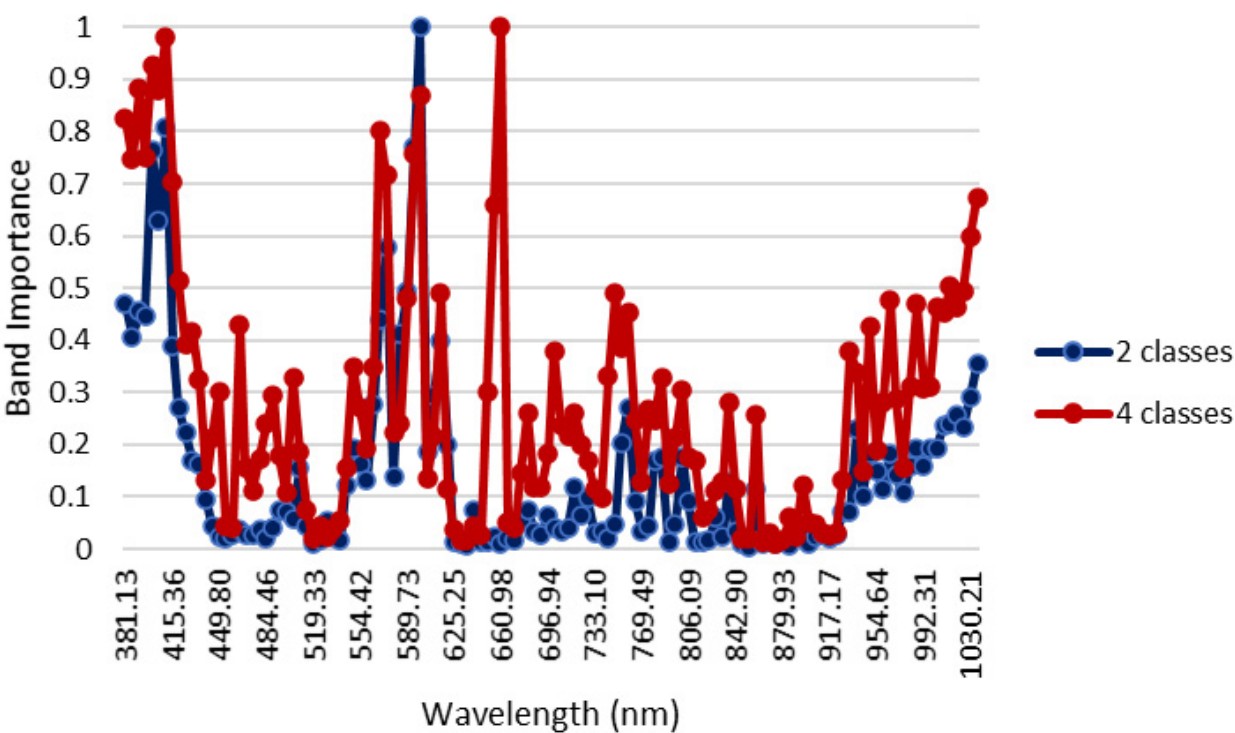

**Figure 15.** Band importance determined by XGBoost on the maximum tuberization phenological stage.

## 4. Discussion

The results indicate that even using a small subset of pixels, taken at random from the hyperspectral images, it is possible to obtain good classification accuracies for detecting and estimating water stress in potato crops. The results also indicate that as early as one day after the onset of the stress in the tubers differentiation phenological stage and on the same day of the onset of the stress in the maximum tuberization water stress can be detected and measured. Other researchers like [33] also found that hyperspectral imaging could be useful to detect water supply conditions of leafy vegetables growing under greenhouse, using modified partial least square regression algorithm, trained to classify different levels of leaf water potential, obtaining a correlation coefficient of 0.826. In this sense, hyperspectral imaging could become a useful tool for the design of precision irrigation systems that allow optimizing the use of water in crops such as potatoes, although it is necessary to develop more studies in real conditions of commercial cultivation.

It was evident that over all classification tasks and phenological stages XGBoost provides excellent classification accuracies alone or in combination with majority voting, followed closely by random forest. Random forest and XGBoost also provide a direct measure of band importance to detect and estimate water stress. In this case, XGBoost seems to better use the whole spectral signature of the canopy, while RF uses a reduced subset of bands. Although the SVM algorithm did not show the best results in this study, the authors of [34] reported promising results when using this algorithm (R = 0.7684) in combination with the Kullback–Leibler divergence (KLD) dimensionality reduction method to select the most relevant bands of hyperspectral images, in the detection of moisture content in maize leaves at the seedling stage. For future experiments, it may be useful to evaluate some combinations of algorithms that have proven to be efficient in the detection of relative water content in leaves, from remote hyperspectral sensing, as reported by [35] who used artificial neural networks (ANN) after selecting the most important bands through partial least squares regression (PLSR), improving the performance of ANN alone.

CNN is a deep learning neural network algorithm that extracts features from images. However, despite being the deep learning neural network most used to analyze images [10], its classification performance was lower than RF and XGBoost, and only by using majority

voting, it was possible to improve its performance to classify all image pixels. This is probably because CNN exploits the spatial structure of the images (such as edges) and not the spectral signature of the images. In this case, the canopy consists of mostly leaves with no spatial clues related to water stress.

Our results indicate that using machine learning and spectral images constitute a phenotyping tool useful to detect and estimate water stress in potato plants, which can also be used in processes of genetic improvement, by choosing those phenotypes that better resist water stress. The reflectance images obtained may be sensitive to the physiological and biochemical changes of the substances and pigments that are degraded and mobilized due to water stress.

## 5. Conclusions

This work shows that detection of water stress, as well as estimation of the water stress level, is possible with good accuracy incremented on the whole canopy, using majority voting at the tubers differentiation and maximum rate of tuberization phenological stages. In particular, the classification results are more accurate and available from the first day of stress for both the tubers differentiation and maximum rate of tuberization phenological stages. Extreme gradient boost performed best overall phenological stages and classification tasks, followed by random decision forests. XGBoost and RF also provide a measure of the importance of each band to detect or estimate water stress in potato crops. In the case of RF, these bands are the violet, red edge, and some specific NIR bands, while in the case of XGBoost it includes some additional bands in the visible (green, yellow, red) and NIR, exploiting better the spectral signature.

These results could lead to the use of more specific normalized water indexes for water stress detection and estimation in potato crops using these machine learning algorithms. However, they are not intended to be used by producers, since this research work was conducted under greenhouse conditions. In this sense, these results are an important basis for further research considering actual potato crop field conditions and cultural practices. It will allow to design advanced tools for early detection of water stress, increasing the efficiency in the application of irrigation.

**Author Contributions:** Conceptualization, E.A.S.-A. and G.A.G.-V.; Methodology, E.A.S.-A.; Software, J.M.D.-C.; Validation, E.A.S.-A.; Formal analysis, L.M.T.-D. and O.D.O.-P.; Investigation, J.M.D.-C. and E.A.S.-A.; Resources, A.M.C.-M.; Data curation, L.M.T.-D. and O.D.O.-P.; Writing—original draft, J.M.D.-C.; Writing—review & editing, E.A.S.-A., G.A.G.-V., L.M.T.-D., O.D.O.-P. and A.M.C.-M. Visualization, L.M.T.-D.; Supervision, A.M.C.-M.; Project administration, A.M.C.-M.; Funding acquisition, A.M.C.-M.; Resources, G.A.G.-V. and A.M.C.-M. All authors have read and agreed to the published version of the manuscript.

**Funding:** This research was funded by the Science, Technology, and Innovation Fund of the General Royalty System, administered by the National Financing Fund for Science, Technology and Innovation Francisco José de Caldas, the Colombia BIO Program, the government of Cundinamarca—Colombia and the Ministry of Science, Technology, and Innovation (MINCIENCIAS), and Corporación Colombiana de Investigación Agropecuaria (AGROSAVIA).

**Institutional Review Board Statement:** Not applicable.

**Acknowledgments:** This work is part of a larger project in Agrosavia called Agroclimatic Information System for potato (*Solanum tuberosum* L.) crops within productive regions in Cundinamarca (SIAP in Spanish). We thank Jose Alfredo Molina Varón for his contribution with the experimental setup and the adaptation of measurement equipment, and Jhon Mauricio Estupiñán Casallas for his collaboration in the assembly of the irrigation systems.

**Conflicts of Interest:** The authors declare no conflict of interest.

## Appendix A

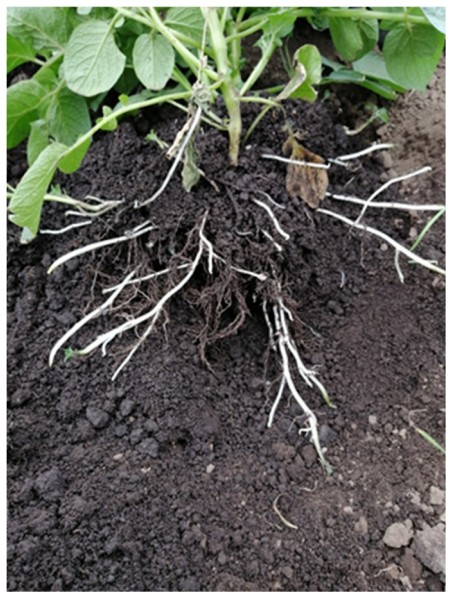 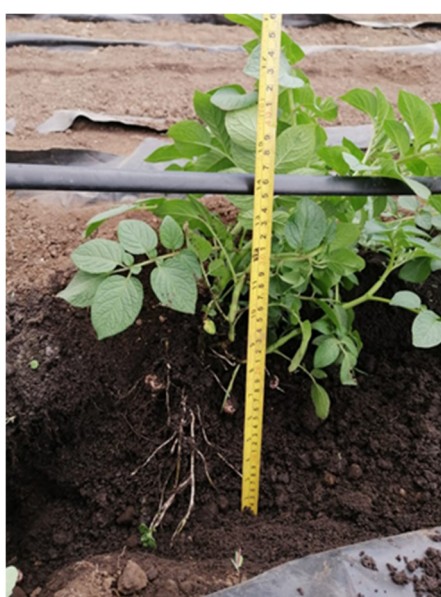

**Figure A1.** Tubers differentiation stage: left and right photographs show the development of stolons: at the apex, "hook" and "matchstick" forms, these are morphological changes in the stolons in the tuber differentiation process.

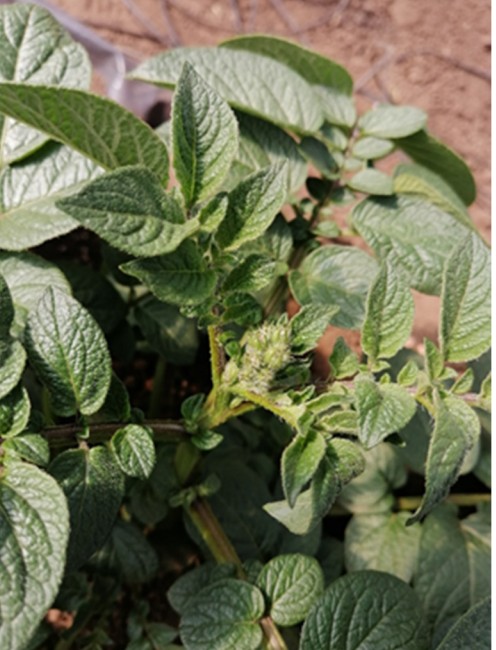 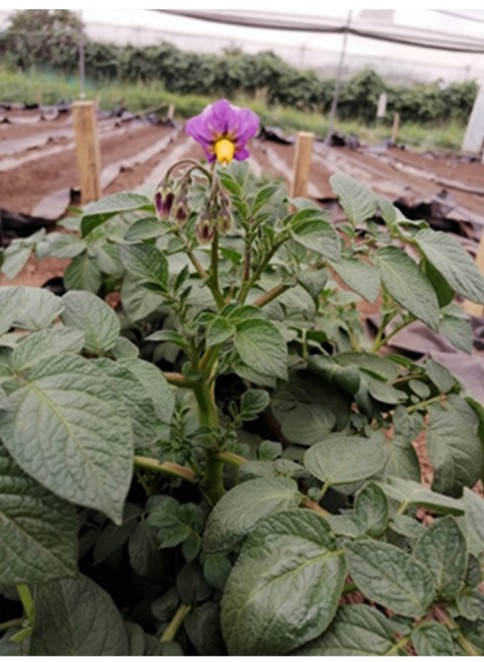

**Figure A2.** Maximum tuberization stage: left and right photographs show the development stage of flowering. Gómez et al. report that the stage of maximum tuberization and beginning of filling coincides with flowering.

## Appendix B

**Table A1.** Four Classes, Tubers Differentiation.

| RF | | | |
|---|---|---|---|
| 63,977.80 | 246.00 | 340.00 | 1436.20 |
| 6514.60 | 10,013.60 | 38.60 | 1433.20 |
| 4497.20 | 2.40 | 7114.20 | 386.20 |
| 8758.40 | 393.20 | 89.80 | 22,758.60 |
| **SVM** | | | |
| 59,562.00 | 538.80 | 166.20 | 5733.00 |
| 13,379.40 | 1381.00 | 37.80 | 3201.80 |
| 9828.60 | 58.00 | 223.20 | 1890.20 |
| 21,264.40 | 388.00 | 47.60 | 10,300.00 |
| **CNN** | | | |
| 52,467.80 | 4374.40 | 1091.40 | 8066.40 |
| 7442.60 | 7230.20 | 79.40 | 3247.80 |
| 4198.00 | 3643.40 | 2123.40 | 2035.20 |
| 9497.20 | 2412.60 | 592.40 | 19,497.80 |
| **MLP** | | | |
| 53,304.20 | 3697.40 | 668.80 | 8329.60 |
| 10,808.20 | 4085.00 | 80.00 | 3026.80 |
| 5405.00 | 2299.80 | 1266.80 | 3028.40 |
| 17,965.20 | 1482.40 | 587.60 | 11,964.80 |
| **XGBoost** | | | |
| 329,137.00 | 90.00 | 69.00 | 704.00 |
| 4314.00 | 85,073.00 | 14.00 | 599.00 |
| 416.00 | 5.00 | 59,409.00 | 170.00 |
| 2869.00 | 32.00 | 25.00 | 157,074.00 |
| **Ada Boost** | | | |
| 51,866.40 | 2866.40 | 3189.60 | 8077.60 |
| 12,534.60 | 2287.20 | 498.00 | 2680.20 |
| 7042.40 | 114.80 | 3204.60 | 1638.20 |
| 18,669.00 | 1172.80 | 1264.00 | 10,894.20 |

**Table A2.** Four Classes, Maximum Tuberization.

| RF | | | |
|---|---|---|---|
| 56,866.20 | 51.20 | 995.00 | 87.60 |
| 1690.80 | 11,658.00 | 618.00 | 33.20 |
| 3996.60 | 539.40 | 19,275.80 | 188.20 |
| 1678.60 | 2.40 | 1060.40 | 5258.60 |
| **SVM** | | | |
| 52,848.20 | 1664.60 | 2907.80 | 579.40 |
| 3598.00 | 8849.20 | 1474.20 | 78.60 |
| 10,426.40 | 2783.60 | 9784.40 | 1005.60 |
| 2296.80 | 165.60 | 2138.20 | 3399.40 |
| **CNN** | | | |
| 50,919.00 | 2418.20 | 3942.80 | 720.00 |
| 1010.40 | 8773.20 | 3995.00 | 221.40 |
| 3454.80 | 299.40 | 18,737.60 | 1508.20 |
| 613.00 | 188.00 | 2661.00 | 4538.00 |

**Table A2.** *Cont.*

| MLP | | | |
|---|---|---|---|
| 56,069.80 | 626.20 | 690.80 | 613.20 |
| 5170.00 | 7803.20 | 942.60 | 84.20 |
| 14,354.60 | 992.60 | 7957.60 | 695.20 |
| 3486.80 | 20.60 | 895.00 | 3597.60 |
| XGBoost | | | |
| 289,738.00 | 150.00 | 91.00 | 21.00 |
| 520.00 | 69,382.00 | 95.00 | 3.00 |
| 291.00 | 26.00 | 119,683.00 | 0.00 |
| 131.00 | 3.00 | 8.00 | 39,858.00 |
| Ada Boost | | | |
| 48,421.80 | 3685.40 | 4595.60 | 1297.20 |
| 2715.60 | 8295.00 | 2920.80 | 68.60 |
| 6624.40 | 3534.20 | 11,884.60 | 1956.80 |
| 1909.00 | 104.80 | 2486.40 | 3499.80 |

**Table A3.** Two Classes, Tubers Differentiation.

| RF | | CNN | | XGBoost | |
|---|---|---|---|---|---|
| 58,628.0 | 7372.0 | 47,915.2 | 18,084.8 | 323,446.0 | 6554.0 |
| 9372.0 | 52,628.0 | 19,694.0 | 42,306.0 | 5475.0 | 304,525.0 |
| SVM | | MLP | | Ada Boost | |
| 43,555.4 | 22,444.6 | 47,570.6 | 18,429.4 | 46,091.2 | 19,908.8 |
| 23,343.4 | 38,656.6 | 24,596.0 | 37,404.0 | 18,315.6 | 43,684.4 |

**Table A4.** Two Classes, Maximum Tuberization.

| RF | | CNN | | XGBoost | |
|---|---|---|---|---|---|
| 54,926.80 | 3073.20 | 52,422.00 | 5578.00 | 288,103.00 | 1897.00 |
| 5220.20 | 40,779.80 | 10,624.40 | 35,375.60 | 1363.00 | 228,637.00 |
| SVM | | MLP | | Ada Boost | |
| 48,065.40 | 9934.60 | 54,502.60 | 3497.40 | 49,767.60 | 8232.40 |
| 8949.60 | 37,050.40 | 16,840.80 | 29,159.20 | 6870.40 | 39,129.60 |

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
