# Peer review of "Estimation of Water Stress in Potato Plants Using Hyperspectral Imagery and Machine Learning Algorithms"

_horticulturae, doi:10.3390/horticulturae7070176_

Round 1
Reviewer 1 Report
This paper presents a study for estimating water stress in potato plants using hyper spectral images. The paper is well organized and easy to follow. Results are well presented and analyzed.
Specific comments:
- Selecting randomly pixels from images to reduce the problem complexity is not very common. This neglects spatial dependencies, and hence can lead to overfitting issues. A better strategy could be to do data augmentation by extracting patches or sub-images from the original data.
- In the same sense, the used CNN is not very sophisticated. In combination with the previously cited issue, the obtained low performance can be explained. A further investigation should be performed.
- Please clarify how the used data is fairly distributed over the different classes for each experiment.
- Any conclusion about the spectral range leading to the best classification performance? Addressing this issue could give more impact to the presented study.
Author Response
We thank the reviewer for the comments on the manuscript that help us improve it. We answer each one of the comments here:
- Selecting randomly pixels from images to reduce the problem complexity is not very common. This neglects spatial dependencies, and hence can lead to overfitting issues. A better strategy could be to do data augmentation by extracting patches or sub-images from the original data.
R: Only CNN uses neighboring information and we provide a window around each pixel for CNN. The rest of machine learning algorithms implemented do not use neighboring information, only the spectral signature of each pixel. We select random pixels due to memory and computational time overhead. Random pixels do select uniformly pixels from all the image, the results show that random sampling works. There is no need for data augmentation, we have too many pixels with 128 bands to choose from. The results indicate that random sampling does work here, as shown by XGBoost and Random Forest cross-validation results (showing no overfitting) and the results of classifying the whole images using the trained classifiers show excellent results too as indicated in Figures 10 and 11, majority voting also provides excellent results.
- In the same sense, the used CNN is not very sophisticated. In combination with the previously cited issue, the obtained low performance can be explained. A further investigation should be performed.
R: Our implementation of CNN is not very sophisticated, but this design has work well in other datasets, better than RF. As indicated in the discussion section, we believe the not so good results of CNN comes from the fact that the images spatial information is low and the CNN implementation does not fully exploit the spectral information, which explains the results.
- Please clarify how the used data is fairly distributed over the different classes for each experiment.
R: We added a comment at the end of Section 2.2 to clarify this (see lines 147-148).
- Any conclusion about the spectral range leading to the best classification performance? Addressing this issue could give more impact to the presented study.
R: In the conclusion section we specified which bands lead to better classification performance.

Reviewer 2 Report
In general, the paper needs to speak to the average potato farmer and help understand and improve yields

Author Response
We thank the reviewer for the comments on the manuscript that help us improve it. We answer each one of the comments here:
In general, the paper needs to speak to the average potato farmer and help understand and improve yields
R: See the new paragraph in the conclusions addressing this issue (lines 367-373).
- (L85-86) How were the measurements of tuber differentiation and tuber filling accomplished?
R: The images were included in Appendix A and description of each stage of development were included in the manuscript (lines 88-90).
- A flat-field transformation could be considered also.
R: There is plenty of data, no need to augment the data with a flat-field transformation.
- I don’t care for the term “own source” is there a replacement suggestion from the journal?
R: We eliminated the term “own source” in all figures.
- (67) Which vegetation Indices? Should be a list.
R: See lines 67-69.
- (116) image, not imagery.
R: Thank you, we corrected this.
- (134) valuable image. SAVI brings out variability.
R: Thank you.
- (149-176) lots of advanced concepts here with only minimum definition or explanation (AB, XGBoost, SVM, RELU, MLP. Could be explained better.
R: We added a small description of the machine learning algorithms employed.
- (188) needs more complete discussion of the over-fitting issue.
R: We added some comments on the overfitting issue.
- The classification accuracy discussion is a bit abstract. Reducing the process to a 6 or 7 digit fraction fails to really communicate the hard work accomplished here. Maybe some type of measurement of the biomass of the potato yield?
R: Biomass variables were measured; however, these figures will be presented in another paper. This new paper will relate the growth parameters with the physiology of stress. This work is about water stress in potato plants and not potato yield. In another work we will address the biochemistry of the plants under water stress.
- What’s the point of the band importance? Typically, such a exercise is used to reduce the important input to a regression process. The authors could consider developing this concept further.
R: We are not interested in regression or dimension reduction in this paper; we are interested in identifying bands that could help the estimation and detection of water stress in potato. These results could help the choice of bands in multispectral cameras or identify vegetation indexes. We added some comments on this on the manuscript.
- Would some images of the various classes of tubers be possible? It would help communicate to non-farmers.
R: The images and description of each stage of development were included in the manuscript.
- This paper is an excellent application of advanced machine learning and hyperspectral imagery. The use of advanced machine learning is a bit abstract though. I believe that incorporation of a classic “confusion matrix would be helpful to us old timers.
R: There are 24 confusion matrices, we added them to the Appendix B, we cannot put them in the manuscript body.
- Also the band importance could be extended into a partial least squares type of application.
R: That´s a possibility for the readers and ourselves in the future, but in this paper, we are interested on band importance for identifying bands that could lead to better multispectral cameras or better vegetation indexes for water stress detection, in combination with machine learning.
- In general, the paper needs to speak to the average potato farmer and help understand and improve yields.
R: Results presented in this paper are not intended to be used by potato farmers, since this research work was conducted under greenhouse conditions. Further research is needed in order to incorporate actual field conditions and practices. It was explained at the end of the conclusions section (lines 367-373).

Round 2
Reviewer 1 Report
The authors addressed my previous comments.
Author Response
Thank You!